# Non-Invasive Evaluation of Kidney Elasticity and Viscosity in a Healthy Cohort

**DOI:** 10.3390/biomedicines10112859

**Published:** 2022-11-08

**Authors:** Felix-Mihai Maralescu, Felix Bende, Ioan Sporea, Alina Popescu, Roxana Sirli, Adalbert Schiller, Ligia Petrica, Bogdan Miutescu, Andreea Borlea, Alexandru Popa, Madalina Bodea, Flaviu Bob

**Affiliations:** 1Division of Nephrology, Department of Internal Medicine II, “Victor Babeș” University of Medicine and Pharmacy, 300041 Timișoara, Romania; 2Centre for Molecular Research in Nephrology and Vascular Disease, “Victor Babeș” University of Medicine and Pharmacy, 300041 Timișoara, Romania; 3Department of Gastroenterology and Hepatology, “Victor Babeș” University of Medicine and Pharmacy, 300041 Timișoara, Romania; 4Advanced Regional Research Center in Gastroenterology and Hepatology, “Victor Babeș” University of Medicine and Pharmacy, 300041 Timișoara, Romania; 5Division of Endocrinology, Department of Internal Medicine II, “Victor Babeș” University of Medicine and Pharmacy, E. Murgu Square, Nr. 2, 300041 Timisoara, Romania

**Keywords:** 2D-SWE PLUS, elasticity, Vi PLUS, inflammation, chronic kidney disease

## Abstract

**Introduction**: There is currently a lack of published data on kidney elasticity and viscosity. Non-invasive techniques, such as two-dimensional shear-wave elastography (2D-SWE PLUS) and viscosity plane-wave ultrasound (Vi PLUS), have surfaced as new detection methods, which, thanks to efficient processing software, are expected to improve renal stiffness and viscosity measurements. This study aims to be the first one to assess the normal range values in normal renal function subjects and to investigate the factors that impact them. **Methods**: We conducted a cross-sectional study employing 50 participants (29 women and 21 men) with a mean age of 42.22 ± 13.17, a mean estimated glomerular filtration rate (eGFR) of 97.12 ± 11 mL/min/1.73 m^2^, a mean kidney length of 10.16 ± 0.66 cm, and a mean body mass index (BMI) of 24.24 ± 3.98. With a C6-1X convex transducer and the Ultra-FastTM software available on the Hologic Aixplorer Mach 30 ultrasound system, we acquired five measurements of renal cortical stiffness and viscosity (achieved from five distinct images in the middle part of the subcapsular cortex) from each kidney. The ten measurements’ median values correlated with the participant’s demographical, biological, and clinical parameters. **Results**: The mean kidney elasticity was 31.88 ± 2.89 kiloPascal (kPa), and the mean viscosity was 2.44 ± 0.57 Pascal.second (Pa.s) for a mean measurement depth 4.58 ± 1.02 cm. Renal stiffness seemed to be influenced by age (r = −0.7047, *p* < 0.0001), the measurement depth (r = −0.3776, *p* = 0.0075), and eGFR (r = 0.6101, *p* < 0.0001) but not by BMI (r = −0.2150, *p* = 0.1338), while viscosity appeared to be impacted by age (r = −0.4251, *p* = 0.0021), eGFR (r = 0.4057, *p* = 0.0038), the measurement depth (r = −0.4642, *p* = 0.0008), and BMI (r = −0.3676, *p* = 0.0086). The results of the one-way ANOVA used to test the differences in the variables among the three age sub-groups are statistically significant for both 2D-SWE PLUS (*p* < 0.001) and Vi PLUS (*p* = 0.015). The method found good intra-operator reproducibility for the 2D-SWE PLUS measurements, with an ICC of 0.8365 and a 95% CI of 0.7512 to 0.8990, and for the Vi PLUS measurements, with an ICC of 0.9 and a 95% CI of 0.8515 to 0.9397. **Conclusions**: Renal stiffness and viscosity screening may become an efficacious, low-cost way to gather supplemental diagnostic data from patients with chronic kidney disease (CKD). The findings demonstrate that these non-invasive methods are highly feasible and not influenced by gender and that their values correlate with renal function and decrease with age progression. Nevertheless, more research is required to ascertain their place in clinical practice.

## 1. Introduction

CKD is becoming more common, and its development is linked to a rise in death rates and healthcare costs. [1] Therefore, detecting CKD sooner and effectively tracking the condition’s advancement using a wide range of indicators are essential.

Classic sonography is utilized to analyze the kidneys, although it only offers a few quantifiable metrics of renal length and parenchyma width, which both reduce as the CKD stage advances. Even though it has been found to have limited diagnostic value in the assessment of CKD, it does provide several pieces of non-quantifiable information regarding the echogenicity of the renal cortex as it improves with fibrosis progression in the later stages of CKD [2]. Even after a previous study attempted to analyze renal parenchyma echogenicity with tools that analyze mean pixel resolution, the detection rate of the traditional ultrasound was still relatively low [3].

Elasticity is described as a property of tissues that enables them to change shape after a load and then regain their original form, with Ophir et al. 1991 [4] being the first one to propose elastography as a method for determining biological tissue elasticity. In recent years, elastography has begun to be utilized as a non-invasive method for the evaluation of a variety of renal pathologies, with the primary goal of diagnosing and monitoring the progression of CKD. The significance of using this approach may lie in its high probability of detecting renal injury in the initial stages of CKD, when numerous renal markers, including the estimated glomerular filtration rate (eGFR) and the albumin-to-creatinine ratio, are still within safe values unless the kidney damage is not so severe [5].

The first and most commonly used technique for evaluating renal stiffness was transient elastography (TE) [6,7,8,9], followed by virtual touch quantification (VTQ) [10,11,12,13,14] and 2D-SWE [15,16]. The findings of Barr et al. in 2020 [17] demonstrate that, up until now, these renal elastography machines did not produce precise shear-wave displacement graphs, and the authors drew the assumption that higher processing systems are required to obtain more precise kidney stiffness data from a non-invasive imaging machine.

Recently, US manufacturers provided better imaging methods that claim to be capable of measuring more specific shear-wave displacement curves in the kidney while at the same time measuring the dispersion of shear-wave characteristics, which can be utilized to estimate viscosity (a metric for shearing motion resistance). Instead of rapid deformation, slow deformation causes tissues to migrate, and necro-inflammation-related alterations are thought to affect how shear waves propagate (viscosity) [18]. Supersonic Imagine created the original 2D-SWE method, and numerous studies and meta-analyses have supported its usefulness in fibrosis assessment [19,20,21,22], but, currently, there are only a few research articles employing the new 2D-SWE PLUS and Vi PLUS technologies [23,24,25].

The purpose of this study was to assess the feasibility and performance of these new ultrasound-based techniques embedded in the new Hologic SuperSonic Mach 30 system (Aixplorer, Supersonic Imagine, Aix-en-Provence, France). To distinguish between normal and pathological cases, it is first necessary to establish the normal kidney elasticity and viscosity values in order to differentiate the factors that influence them, as well as their variability in healthy subjects.

## 2. Materials and Methods

### 2.1. Study Population

In a tertiary department of nephrology over a seven-month period (March 2022 to September 2022), a monocentric, cross-sectional study was conducted. Using 2D-SWE PLUS and Vi PLUS from the new Hologic Aixplorer Mach 30 ultrasound system(Aixplorer, Supersonic Imagine, Aix-en-Provence, France), elastography-based measurements were carried out on all subjects during the same session by a single operator with three years of experience in kidney elastography (F.-M.M). The study was carried out in accordance with the World Medical Association Declaration of Helsinki, amended in 2000, Edinburgh, and it was authorized by our university’s ethics committee for research and institutional review board (number 41/4 March 2022). Before enrolling in the trial, each patient gave their written informed consent.

Fifty healthy subjects consisting of hospital employees (doctors, nurses, students, and auxiliary staff) took part in the study following the confirmation of informed consent. We collected the following information from each participant: age, sex, height, weight, BMI, and serum creatinine. Prior to the elastography measurements, we performed a urine dipstick test on each participant. The eligible participants for the control group were those who were aged over 18, well-hydrated (with affirmative normal urine output and no mouth dryness or skin turgor), and without obesity (BMI lower than 30) and who had normal renal function (eGFR > 60 mL/min/1.73 m^2^), no illness that might affect the findings (high blood pressure, diabetes, neoplasms, heart or liver diseases), and regular kidneys on conventional ultrasonography. We measured the renal parenchyma thickness in all participants, and those with a renal parenchyma thickness under 10 mm were eliminated. The exclusion criteria were hydronephrosis, autosomal dominant polycystic disease, pregnancy, clinical signs of upper urinary tract infection, and refusal to provide informed consent.

### 2.2. Elastography Using 2D-SWE PLUS and Vi PLUS

Using a C6-1X convex probe, Hologic Aixplorer Mach 30 ultrasound software was employed to evaluate the 2D-SWE PLUS measurements. The machine’s operating system was utilized to calculate the Young’s modulus of the region of interest (ROI), using the equation E = ρ*cs2, where E represents the elasticity of the tissue expressed in kPa, ρ represents the tissue density measured in kg/m^3^, and finally cs represents the shear-wave velocity measured in m/s [26]. A quantifiable image of tissue stiffness is depicted using imaging techniques. The color progresses from blue to yellow to red, representing the Young’s modulus values ranging from zero to more than fifty kPa [27].

After the subject emptied their bladder, measurements were taken in the central section of the kidney right under the subcapsular cortex, while they were in the lateral decubitus position. The equipment software produced data for each measurement (the ROI is pre-established at 10mm by the kidney software program and is displayed on the screen as a Q-box). The “Depth” parameter represents the distance from the skin to the ROI, which is used for measuring the 2D-SWE PLUS and Vi PLUS values (Figure 1).

Operations were carried out with the participant in neutral respiratory apnea. We collected 5 valid readings of renal stiffness and viscosity from each participant’s kidney (5 different images in the middle portion of the subcapsular cortex), blinded to the clinical and biological data (every person included presented in the last 6 months with normal serum creatinine and a normal urinary exam), with a mean time of examination of approximately 10 min. The subjects’ demographic and clinical parameters were correlated with the mean values of the ten non-invasive imaging measurements.

The shear-wave measuring box was positioned in the center of the renal parenchyma, immediately underneath the renal cortex, after choosing the most appropriate window using a traditional ultrasound scan and achieving a favorable image (a representation of the entire kidney in a single frame, without superimposed images and the outer renal cortex having a fine and uniform echotexture).

The success rate of performing elastography in native healthy kidneys was 100%; however, it is true that, in patients with CKD (excluded from this pilot study), who sometimes have a thinner renal parenchyma, it is quite difficult to differentiate between the cortex and medulla and to perform the measurements exclusively in the cortex. 

Vi PLUS can now be utilized to display information about tissue shear-wave dispersion by performing an examination of shear-wave propagation velocity at varying wavelengths. In a color-coded chart, the degree of the shear-wave speed shift between frequencies is qualitatively shown, and it is provided mathematically in Pa.s across a value range. Vi PLUS was utilized in tandem with the 2D-SWE PLUS mode, and both followed the same technique.

### 2.3. Statistical Analysis

For the statistical analysis, MedCalc software Version 19.4(MedCalc Software Corp., Brunswick, ME, USA) and Excel from Microsoft Office 2020 for Windows were utilized. Descriptive statistics were used to analyze the demographic and anthropometric findings. The Kolmogorov–Smirnov test was used to assess the distribution of the numeric values. Correlations between variables are expressed using the Pearson or Spearman correlation coefficients, with a “p“ of under 0.05 being considered statistically significant. Univariate and multivariate regression models were used in order to determine which factors influence 2D-SWE PLUS and Vi PLUS values. One-way ANOVA was used to test the difference in variables among three age sub-groups, and box-and-whisker plots were provided afterward to better understand the distinctions between them.

## 3. Results

A total of 29 women and 21 men with a mean age of 42.22 ± 13.17, a mean estimated glomerular filtration rate (eGFR) of 97.12 ± 11 mL/min/1.73 m^2^, a mean kidney length of 10.16 ± 0.66 cm, and a mean BMI of 24.24 ± 3.98 kg/m^2^ were included in our analysis (Table 1).

For the whole group, the mean kidney elasticity was 31.88 ± 2.89 kPa, and the mean viscosity was 2.44 ± 0.57 Pa.s for a mean measurement depth of 4.58 ± 1.02cm. eGFR seemed to influence both the 2D-SWE PLUS (r = 0.6101, *p* < 0.0001) and Vi Plus (r = 0.4057, *p* = 0.0038) values. We also found a negative correlation between age and eGFR (r = −0.8521, *p* < 0.0001) and a positive correlation between the median measurements of the 2D-SWE PLUS and Vi PLUS values with an r = 0.2892, *p* = 0.0417. No statistically significant differences between genders were noticed: the mean kidney stiffness values in men were 32.03 ± 2.84 kPa, slightly higher than the mean stiffness values in women at 31.08 ± 2.5 kPa (*p* = 0.2185) (Figure 2), while the mean viscosity values in men were 2.49 ± 0.47 Pa.s, a bit lower than that in women at 2.52 ± 0.79 Pa.s (*p* = 0.8964) (Figure 3).

Age seemed to influence both renal stiffness (r = −0.7047, *p* < 0.0001) (Table 2 and Figure 4) and viscosity (r= −0.4251, *p* = 0.0021) (Table 3 and Figure 5), but, regarding BMI, we found no correlation with renal stiffness (r = −0.2150, *p* = 0.1338), only with viscosity (r= −0.3676, *p* = 0.0086). The results of the one-way ANOVA used to test the differences in variables among the three age sub-groups are statistically significant for both 2D-SWE PLUS (*p* < 0.001) and Vi PLUS (*p* = 0.015).

We also found a positive correlation between the mean measurement depths and the 2D-SWE PLUS values (r = −0.3776, *p* = 0.0075), as well as with the Vi PLUS measurements (r= −0.4642, *p* = 0.0008). The method found good intra-operator reproducibility for the 2D-SWE PLUS measurements, with an ICC of 0.8365 and a 95% CI of 0.7512 to 0.8990, and for Vi PLUS, with an ICC of 0.9 and a 95% CI of 0.8515 to 0.9397. 

We discovered that eGFR, the median measurement depths, the median viscosity, and age influenced the 2D-SWE PLUS and Vi PLUS values in the univariate regression, but when they were included in the multivariate analysis, no statistically significant model was found.

## 4. Discussion

In reality, determining tissue mechanical properties is impossible, but to fully comprehend these new US-based parameters, it is crucial to assess the baseline data from healthy kidney subjects of various ages and sexes and to further investigate the variables that affect them. Nonetheless, to the best of our knowledge, no other clinical study has concentrated on identifying the reference values of kidney stiffness and viscosity in healthy kidney subjects by employing these new, non-invasive methods. 

The mean kidney stiffness value was 31.88 ± 2.89 kPa, and the mean viscosity value was 2.44 ± 0.57 Pa.s. As a result, a 2D-SWE PLUS measurement of around 31.88 kPa and a Vi PLUS value of around 2.44 Pa.s are indicative of a healthy kidney free of fibrosis or inflammation. Age appears to have an effect on renal stiffness; deterioration is a physiological process of cellular and organ senescence, and, therefore, it is linked with structural changes in the kidneys. A potential explanation for the decrease in stiffness with advanced age could be the fact that renal blood flow decreases with age due to these structural changes. The mean 2D-SWE PLUS and Vi PLUS measurements decreased with age but were unaffected by the participant’s sex. In addition, we also published a study on kidney transplant recipients evaluated using the same elastography measurements in which we obtained a cut-off value of 27.3 kPa for estimating an eGFR of under 60 mL/min/1.73 m^2^, and this also indicated that, when the disease advances, renal stiffness declines [23].

A decrease in renal stiffness with the progression of CKD has also been shown in other kidney elastography studies [10,13,28,29,30,31,32,33], but there have also been some studies that revealed an increase in stiffness as the disease progressed [11,34,35,36,37]. Despite the fact that renal elastography seems to be a viable technique for tracking the progression of CKD, existing research has revealed substantial differences among such techniques [38]. The high degree of anisotropy of the kidney [39] or modifications in renal blood flow may have an impact on renal stiffness and justify the discrepancies in the outcomes. As CKD advances, reduced stiffness may be caused by a decrease in renal perfusion, which may have a bigger influence on stiffness than kidney fibrosis [40].

The depth of the kidneys was another issue related to renal elastography until now. The placing of the measurement box in native kidneys is a massive obstacle in the non-invasive evaluation of renal stiffness and viscosity, and previous methods have been limited in their capacity to analyze deep tissues. The target anatomical depth of acoustic radiation force impulse-based approaches is reported to be 7 cm [41]. The software of Hologic Mach 30 provided us with a deep penetration mode that was able to detect deeper than 7 cm, offering an improved future option for patients with morbid obesity whose kidneys may exceed the previous limit.

The inflammatory process is essential in the advancement of fibrosis [42]. Vi PLUS collects data on tissue shear-wave dispersion, and these data can then be utilized to calculate viscosity [43]. Our research found a link between Vi PLUS, which decreases with age, and with the measurement depth and BMI. Only a few studies that evaluated the liver used this non-invasive method to measure the viscosity of tissue [24], with Deffieux et al. (2015) [44] being the first. We could indeed theorize that viscosity metrics that illustrate inflammatory states could be beneficial when assessing patients who have suffered acute kidney injury or acute pyelonephritis or when assessing kidney transplant recipients to evaluate acute rejections.

The performance capability of acoustic radiation force impulse for measuring renal parenchymal stiffness was assessed in a meta-analysis by Hwang et al., 2021 [45]. The proportions of technical difficulties and intrasubject correlation coefficients agreed well in native and transplanted kidneys, but the placement of the ROI was a major cause of heterogeneity. Both of the non-invasive measures in our study show good intra-operator agreement.

However, more research on patients with CKD using a kidney biopsy as a reference method is required to learn more about the various factors that actually influence kidney elasticity and viscosity. Nonetheless, the current study included volunteers who had never had kidney disease, and obtaining a biopsy would be inappropriate. Non-invasive evaluations are unlikely to be able to rival the gold standard’s diagnostic power, but their capacity to monitor variations in the parenchymal structure as time passes may be the most plausible and attractive reasons for their use.

The limitations of the current research are the study’s small number of participants, the examination of a small proportion of variables that may influence cortical stiffness and viscosity (lack of the urinary albumin-to-creatinine ratio [46] or not considering vascular changes in the kidney), and the fact that there are no recommendations in the guidelines or quality criteria regarding renal elastography.

## 5. Conclusions

Renal stiffness and viscosity screening may become an efficacious, low-cost way to gather supplemental diagnostic data from patients with CKD. The findings demonstrate that these non-invasive methods are highly feasible and not influenced by gender and that their values correlate with renal function and decrease with age progression. Nevertheless, more research is required to ascertain their place in clinical practice.

## Figures and Tables

**Figure 1 biomedicines-10-02859-f001:**
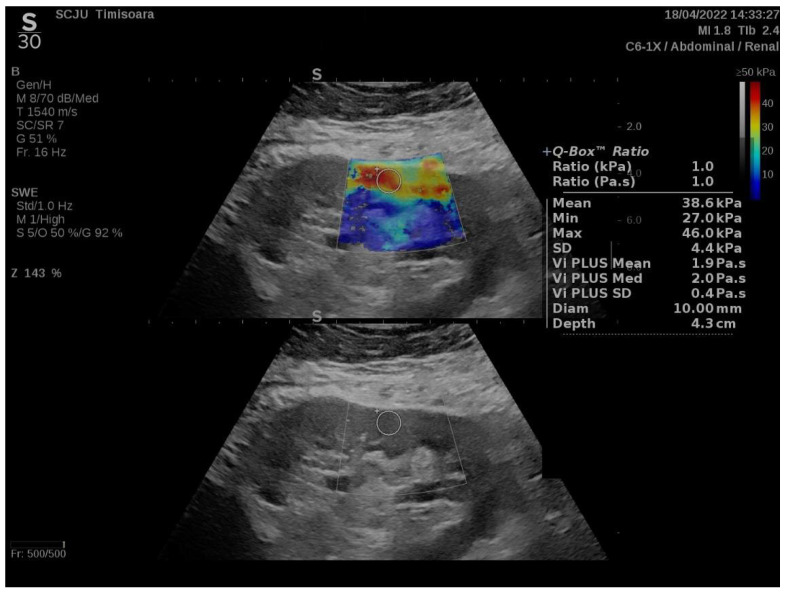
A healthy participant’s kidney viscoelasticity map.

**Figure 2 biomedicines-10-02859-f002:**
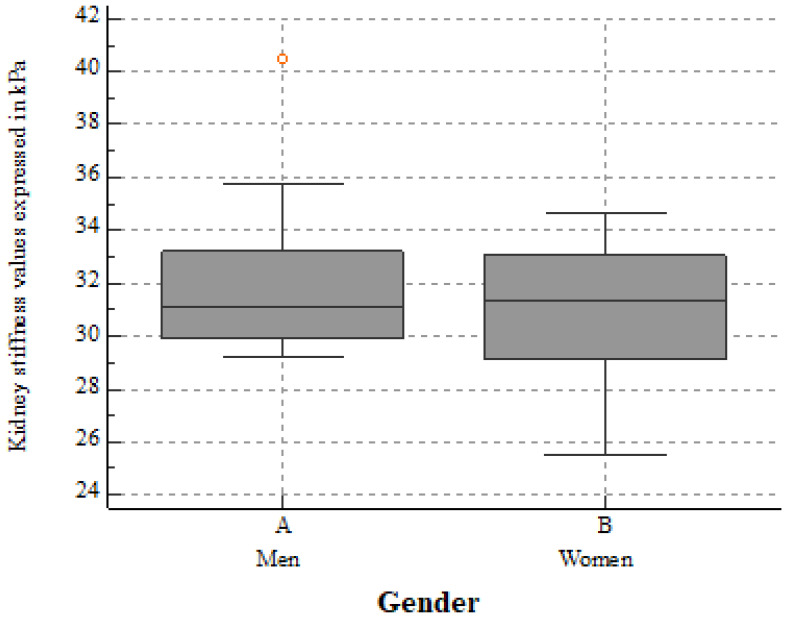
The mean kidney stiffness values in healthy men and women. No significant differences between 2D-SWE PLUS mean values were found (*p* = 0.2185).

**Figure 3 biomedicines-10-02859-f003:**
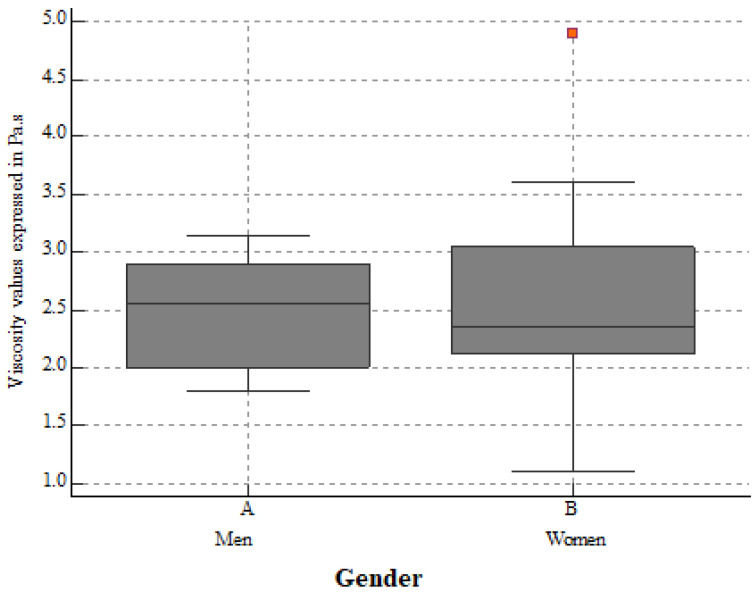
The mean kidney viscosity values in healthy men and women. No significant differences between Vi PLUS mean values were found (*p* = 0.8964).

**Figure 4 biomedicines-10-02859-f004:**
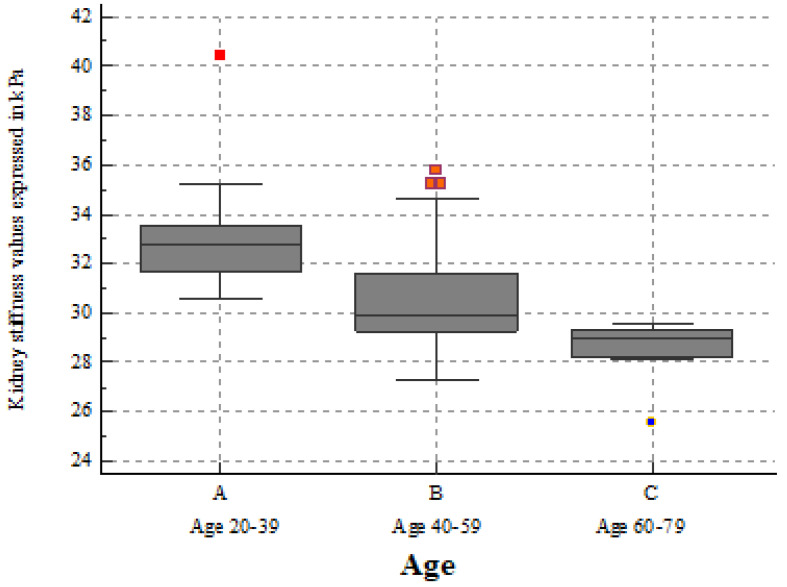
Box-and-whisker distribution plot showing 2D-SWE PLUS values across three different age subgroups. The 2D-SWE PLUS values decreased slightly with age progression.

**Figure 5 biomedicines-10-02859-f005:**
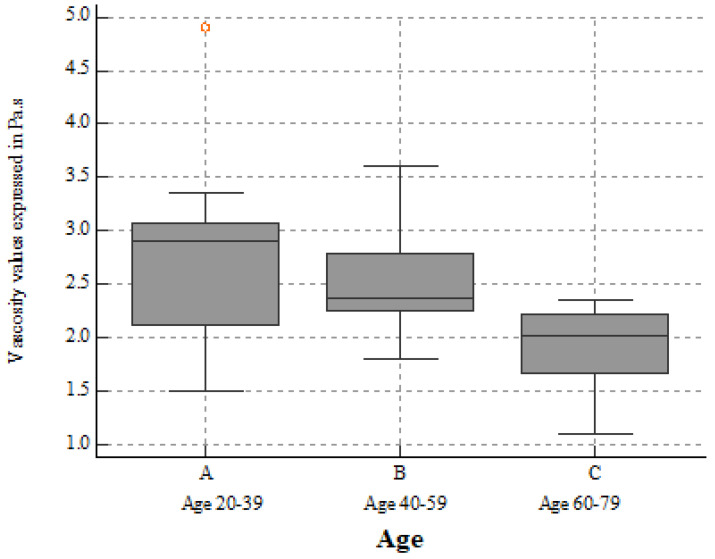
Box-and-whisker distribution plots showing Vi PLUS values across three different age subgroups. Vi PLUS values decreased slightly with age progression.

**Table 1 biomedicines-10-02859-t001:** Baseline characteristics.

Variable	Subjects (n = 50; 29 Women/21 Men)
Age (years)	42.22 ± 13.17
eGFR (mL/min/1.73m^2^)	97.12 ± 11
Kidney length (cm)	10.16 ± 0.66
BMI (kg/m^2^)	24.24 ± 3.98

**Table 2 biomedicines-10-02859-t002:** The differences in variables among the three age sub-groups for 2D-SWE PLUS.

Factor (Age Sub-Group)	n	Mean 2D-SWE PLUS Values (kPa)	SD	Different (*p* < 0.05) from Factor no.
(1) **20–39 years**	n = 23	32.9630	2.0425	(2)(3)
(2) **40–59 years**	n = 18	30.9278	2.5677	(1)(3)
(3) **60–79 years**	n = 9	28.5187	1.2970	(1)(2)

One-way ANOVA was used to test the differences in variables among the three age sub-groups for 2D-SWE PLUS. *p* < 0.001; n = number of participants; SD = standard deviation.

**Table 3 biomedicines-10-02859-t003:** The differences in variables among the three age sub-groups for Vi PLUS.

**Factor** **(Age Sub-Group)**	**n**	**Mean Vi PLUS Values (Pa.s)**	**SD**	**Different (*p* < 0.05) from Factor no.**
(1) **20–39 years**	n = 23	2.6891	0.7274	(3)
(2) **40–59 years**	n = 18	2.5528	0.5524	-
(3) **60–79 years**	n = 9	1.9125	0.4241	(1)

One-way ANOVA was used to test the differences in variables among the three age sub-groups for Vi PLUS. *p* = 0.015; n = number of participants; SD = standard deviation.

## Data Availability

Not applicable.

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
