# Peer review of "Non-Invasive Evaluation of Kidney Elasticity and Viscosity in a Healthy Cohort"

_biomedicines, 2022, doi:10.3390/biomedicines10112859_

Round 1
Reviewer 1 Report
The author conducted a cross-sectional study to elucidate the normal range values of elasticity and viscosity in a cohort with normal renal function. The idea of this study is interesting. However, some issues should be clarified.
Major:
1. The sample size is small to set a normal range reference of renal elasticity and viscosity because the results can’t be generally applied. Therefore, the result can’t answer the main purpose of this manuscript.
2. In the introduction, the authors should mention the detail of the research gap, which this study wants to elucidate.
3. The baseline information of participants should be provided as a separate table.
4. In statistical analysis, just using one-way ANOVA can’t adjust the effect of unbalanced baseline data because this is not a randomized control study. Please consider the multivariable adjusting model.
5. The eGFR should be considered and put into analysis.
Author Response
We would like to thank the reviewer for the suggestions, which have undoubtedly helped us considerably enhance our paper. We have made major modifications and hope that the publishing of this article will encourage additional research on this topic.
- The sample size is small to set a normal range reference of renal elasticity and viscosity because the results can’t be generally applied. Therefore, the result can’t answer the main purpose of this manuscript.
- We agree that the number of participants is relatively small, which is why we addressed this aspect in our study's limitations.
- In the introduction, the authors should mention the detail of the research gap, which this study wants to elucidate.
- The goal of this research was to evaluate the feasibility and performance of these new ultrasound-based approaches in healthy kidneys. We believe that the last paragraph of the introduction is representative of this aspect.
- The baseline information of participants should be provided as a separate table.
We would like to thank the reviewer for this recommendation, we added a table regarding baseline information at the beginning of the results section.
- In statistical analysis, just using one-way ANOVA can’t adjust the effect of unbalanced baseline data because this is not a randomized control study. Please consider the multivariable adjusting model.
We would like to thank the reviewer for this suggestion as well, we performed univariate and then multivariate regression models to see which factors actually influence kidney stiffness and viscosity but no statistically significant models were found.
- The eGFR should be considered and put into analysis.
We would also want to thank the reviewer for the last recommendation. We included GFR in our analysis and performed correlations. All changes have been performed using track changes so that they can be easily followed.
Reviewer 2 Report
The manuscript is a monocentric cross-sectional study in non-invasive ultrasound evaluation of kidney elasticity and viscosity in a healthy cohort. The aims were to assess the normal range values in normal renal function subjects and to investigate the factors that impact them.
Major comments
Data on the patient's demographic and clinical (including laboratory) parameters are missing. Show the data in a table.
Line 132 "Every person included presented in the last 6 months with normal serum creatinine and urinary exam." Lines 104-107 The eligible participants for the control group were the ones with age over 18, well hydrated (with normal urine output, no mouth dryness or skin turgor), without obesity (BMI lower than 30), normal renal function (eGFR>90ml/min/1.72m2), and the exclusion of any illness that might affect the findings (high blood pressure, diabetes, neoplasms, heart or liver diseases), as well as regular kidneys on conventional ultrasonography." Lines 257-260 "Limitations of the current research would be the study’s small number of participants, the examination of a small proportion of variables that may influence cortical stiffness and viscosity (lack of urinary albumin to creatinine ratio [46] or not considering vascular changes in the kidney."
1) Serum creatinine within the normal range does not mean eGFR greater than or equal to 90 mL/min/1.73 m2. Are there different gender ranges for serum creatinine values in your laboratory?
2) Which exact parameters does the urinary exam include in your study? How did you determine the urinary output in the subject?
3) How did you rule out other diseases or comorbidities that could affect the interpretation of the findings/results? By self-reporting or otherwise?
Include Your comments in the manuscript.
Demographic and clinical data on subjects should be presented in the Results, not in the Methods.
Lines 100-101 "...with a mean age of 42.22 ± 13.17, a mean kidney length of 10.16±0.66cm, and a mean BMI of 24.24±3.98 took part in the study..."
All abbreviations used in the text should be given the full name at the first reference in the text. Example is ROI.
Figure (images) 2.-5. are of poor quality and have a blurred appearance. Provide a better ones.
Align the references in the manuscript according to the instructions for authors.
Minor comments
Lines 105-106 ".. normal renal function (eGFR>90ml/min/1.72m2)"
Correct is 1,73 m2.
In several places in the manuscript there is a full stop (.) at the end of the sentence before the parenthesis ([3]). Examples are following lines: 57, 66, 78, 217,220,222, 226, 231,235, 237
In several places in the manuscript, a space is missing between words. Examples are following lines: 168, 219.
There are also redundant full stops in the manuscript. Examples are following lines: 167, 219.
Rename tables 1 and 2, and list the statistic methods used with other clarifications (abbreviations etc.) below the tables. Remove row shading in tables.
Example for title for Table 1. The differences in variables among the three age sub-groups for 2D-SWE PLUS.
Author Response
We would like to thank the reviewer for the suggestions, which have undoubtedly helped us considerably enhance our paper. We have made major modifications and hope that the publishing of this article will encourage additional research on this topic.
Data on the patient's demographic and clinical (including laboratory) parameters are missing. Show the data in a table.
We would like to thank the reviewer for this observation, we included a table at the beginning of the results section.
Line 132 "Every person included presented in the last 6 months with normal serum creatinine and urinary exam." Lines 104-107 The eligible participants for the control group were the ones with age over 18, well hydrated (with normal urine output, no mouth dryness or skin turgor), without obesity (BMI lower than 30), normal renal function (eGFR>90ml/min/1.72m2), and the exclusion of any illness that might affect the findings (high blood pressure, diabetes, neoplasms, heart or liver diseases), as well as regular kidneys on conventional ultrasonography." Lines 257-260 "Limitations of the current research would be the study’s small number of participants, the examination of a small proportion of variables that may influence cortical stiffness and viscosity (lack of urinary albumin to creatinine ratio [46] or not considering vascular changes in the kidney."
1) Serum creatinine within the normal range does not mean eGFR greater than or equal to 90 mL/min/1.73 m2. Are there different gender ranges for serum creatinine values ??in your laboratory?
- We would like to thank the reviewer for this important remark, which we also have modified to “>60ml/min/1,73m2”. We do not have different gender range values for serum creatinine in our laboratory.
2) Which exact parameters does the urinary exam include in your study? How did you determine the urinary output in the subject?
- Prior to the elastography measurements, we performed a urine dipstick test on each participant. Every person included in the study affirmed having a normal urinary output.
3) How did you rule out other diseases or comorbidities that could affect the interpretation of the findings/results? By self-reporting or otherwise?
- Medical data for each participant was collected from the occupational health service at our hospital.
Demographic and clinical data on subjects should be presented in the Results, not in the Methods.
We would like to thank the reviewer for this observation as well. We modified and included the information in the results section.
Lines 100-101 "...with a mean age of 42.22 ± 13.17, a mean kidney length of 10.16±0.66cm, and a mean BMI of 24.24±3.98 took part in the study..."
All abbreviations used in the text should be given the full name at the first reference in the text. Example is ROI
We would like to thank the reviewer for pinpointing this aspect as well, we added the full names for the abbreviations.
Figure (images) 2.-5. are of poor quality and have a blurred appearance. Provide a better ones.
Unfortunately, these are the ones provided by MedCalc Software, which we can not modify.
Lines 105-106 ".. normal renal function (eGFR>90ml/min/1.72m2)" Correct is 1,73 m2.
We would like to thank the reviewer for this observation, we modified the text as well.
In several places in the manuscript there is a full stop (.) at the end of the sentence before the parenthesis ([3]). Examples are following lines: 57, 66, 78, 217,220,222, 226, 231,235, 237 In several places in the manuscript, a space is missing between words. Examples are following lines: 168, 219. There are also redundant full stops in the manuscript. Examples are following lines: 167, 219.
We would like to thank the reviewer for these observations as well, we modified them in the text.
Rename tables 1 and 2, and list the statistic methods used with other clarifications (abbreviations etc.) below the tables. Remove row shading in tables.
Example for title for Table 1. The differences in variables among the three age sub-groups for 2D-SWE PLUS.
We really appreciate the comments, we removed the shading from the tables and modified the titles, and added the abbreviations below as suggested. All changes have been performed using track changes so they can be easily followed.
Round 2
Reviewer 1 Report
acceptable revision
Reviewer 2 Report
The authors have made major improvement following reviewers suggestions and comments.